# Significance of Fermentation in Plant-Based Meat Analogs: A Critical Review of Nutrition, and Safety-Related Aspects

**DOI:** 10.3390/foods12173222

**Published:** 2023-08-27

**Authors:** Hosam Elhalis, Xin Yi See, Raffael Osen, Xin Hui Chin, Yvonne Chow

**Affiliations:** Singapore Institute of Food and Biotechnology Innovation (SIFBI), Agency for Science, Technology and Research (A*STAR), 31 Biopolis Way, Nanos, Singapore 138669, Singapore; h.elhalis@unswalumni.com (H.E.); see_xin_yi@sifbi.a-star.edu.sg (X.Y.S.); raffael_osen@sifbi.a-star.edu.sg (R.O.); chin_xin_hui@sifbi.a-star.edu.sg (X.H.C.)

**Keywords:** meat alternatives, anti-nutritive and digestibility, alternative proteins, starter cultures, sustainable

## Abstract

Plant-based meat analogs have been shown to cause less harm for both human health and the environment compared to real meat, especially processed meat. However, the intense pressure to enhance the sensory qualities of plant-based meat alternatives has caused their nutritional and safety aspects to be overlooked. This paper reviews our current understanding of the nutrition and safety behind plant-based meat alternatives, proposing fermentation as a potential way of overcoming limitations in these aspects. Plant protein blends, fortification, and preservatives have been the main methods for enhancing the nutritional content and stability of plant-based meat alternatives, but concerns that include safety, nutrient deficiencies, low digestibility, high allergenicity, and high costs have been raised in their use. Fermentation with microorganisms such as *Bacillus subtilis*, *Lactiplantibacillus plantarum*, *Neurospora intermedia*, and *Rhizopus oryzae* improves digestibility and reduces allergenicity and antinutritive factors more effectively. At the same time, microbial metabolites can boost the final product’s safety, nutrition, and sensory quality, although some concerns regarding their toxicity remain. Designing a single starter culture or microbial consortium for plant-based meat alternatives can be a novel solution for advancing the health benefits of the final product while still fulfilling the demands of an expanding and sustainable economy.

## 1. Introduction

Meat and meat products represent an essential source of protein, with the global market estimated at over USD 897 billion in 2021 [1,2]. By 2050, the global population is projected to have grown to roughly 9 billion people, which will require at least twice as much protein as is currently produced [3,4]. To meet this demand, meat production must increase, but this increase is severely constrained by scarcity in water and land resources. In addition, the negative effects of the meat industry on the environment and climate change, the rising concern for animal welfare, and the growth of the halal and kosher markets all point to the necessity of meat analogs (also known as meat replacers, meat substitutes, or meat alternatives) for supporting this growing demand [5,6].

Cultured meat, edible insects, mycoprotein, and plant-based proteins are expected to be the main protein sources for formulating meat analogs. Among them, plant-based meat analogs are gaining popularity in the market and have been promoted as a healthy, environmentally friendly, and ethical solution [7]. Plant-based diets are reported to have lower adverse impacts on human health that include lowering blood pressure, and incidences of cardiovascular diseases and diabetes [8,9,10]. Additionally, they are proven to be more eco-friendly and produce less greenhouse gases [11,12]. Tempeh, tofu, and seitan are the first generation of plant-based food products. These products are popular in Asian countries but have lower consumer acceptability in the West [13]. A new generation of items made from plant ingredients that mimic real meat are already in the market and called plant-based meat analogs [14,15]. Plant-based meat analogs are defined as plant proteins that are structured and formulated to mimic the flavor, appearance, texture, and nutritional profile of meat [16,17]. Such products are designed to appear and taste like meat, contain a similar amount of protein, and are generally lower in fats and calories compared to meat products [18,19]. Although considerable effort has been employed to improve the appearance, flavor, and texture [7,20] of these meat analogs, little attention has been given to enhancing their health benefits and improving their safety. Fortifications include vitamins such as riboflavin, thiamine, niacin, cobalamin, and pyridoxin, and minerals such as calcium, zinc, and iron, as well as fiber from vegetables, oat, bamboo, and pea, which are common ingredients that have been used in plant-based meat alternatives. There is little evidence to prove that these additives offer a significant health benefit [21,22,23,24,25]. Additionally, plant-based meat analogs are ultra-processed foods and reported to have anti-nutrients such as trypsin inhibitors, phytates, oligosaccharides, and allergens, which raise health concerns [26,27,28]. This highlights the need for food industries to find solutions to these challenges [7,29].

Fermentation is one of the oldest methods used to preserve perishable foods and improve their safety, nutritional, and sensory quality in an economical and energy-efficient way [30,31]. Fermentation improves the nutritional value by boosting the quantities of vitamins, amino acids, fatty acids, and other bioactive components [32]. As such, fermentation may be a good option for improving the nutritional value of plant-based meat substitutes. Unlike conventional spontaneous fermentation, starter culture technology uses formulators to exert control over fermentation processes and adjust the quality of the final products [33,34,35,36,37]. A starter culture contains a wide range of active microorganisms, including bacteria and fungi, that are added to initiate desirable changes during the product manufacturing process. The use of fermented plant materials and starter culture technology in the production of plant-based meat analogs has received little attention, despite its potential in improving the nutritional value of meat alternatives. This review provides an overview of the current state of scientific knowledge regarding the challenges to the safety and nutritional aspects of plant-based meat analogs and introduces the use of fermentation and starter culture technology as novel strategies for more sustainable and economical ways of producing safe, nutritional plant-based meat analogs.

## 2. Plant-Based Meat Analogs Characterization and the Demand for New Approaches

Plant-based meat alternatives are a broad category of foods that are similar in texture, appearance, flavor, and nutritional quality to real meat products in the human diet. For the purpose of this review, real meats are defined as fresh uncooked whole muscle meat such as boneless steaks, whole, cooked cuts such as beef roast, or fermented meats, such as fermented sausages. Nutrition quality is considered a key driver for consumer acceptance of plant-based meat analogs [7,38]. Table 1 summarizes the nutritional content and illustrates some selected advantages and disadvantages of real meat and plant-based meat alternatives. Details regarding components, formulation, and structuring technologies of plant-based meat analogs and their functions are outside the scope of this review and have been reviewed in depth by other authors [7,39,40,41,42]. However, we briefly highlight relevant points to explain the drawbacks of current processing methods and the need for a novel approach. 

Plant-based meat analogs contain protein, fiber, low levels of saturated fat, and no cholesterol [39,69]. They are created from plant proteins and non-proteins mixed with other additives and subjected to downstream processes to improve their texture, appearance, flavor, and taste [14,15]. Such products are designed to appear and taste like conventional meat-based products. They contain similar concentrations of proteins but are lower in carbohydrates, fats, and calories compared to real meat products [18,19]. Soybeans are the major source of plant protein used but other vegetable proteins such as peas, check beans, cotton seeds, canola seeds, and rape seeds have also been used [14]. Vegetable oils including coconut oil, avocado oil, cocoa butter, corn oil, soybean oil, and sunflower oil are among the most common source of lipids used in meat alternatives [70]. Glucose, fructose, starch, and methyl cellulose are commonly utilized as carbohydrates in meat alternatives [71]. Furthermore, plant-based meat analogs are usually fortified with micronutrients like zinc, iron, folic acid, and vitamins B1, B2, B6, and B12 [72,73]. Encapsulation techniques including emulsion and non-emulsion systems may be applied to improve the delivery of sensitive functional ingredients that may be degraded during processing or storage such as vitamins [74,75,76]. For the microbial safety and stability of the final product, plant-based preservatives with antioxidant and antimicrobial functionalities are added to the meat analogs [76]. For example, carotenoids, tocopherols, spices, and herbs are used as antioxidants, and curcumin, essential oils, and polyphenols are used as antimicrobials [70,77]. Despite the apparent nutritional completeness of these meat alternatives, recent concerns on their nutritional content and safety have been raised. These include the presence of antinutritive factors, food pathogens, and genetically modified ingredients as well as their low digestibility, high allergenicity, and nutrition deficiency. There is thus an urgent need for further improvement in these meat alternatives, as discussed below.

### 2.1. Anti-Nutritional Factors

Many plant proteins contain anti-nutritional factors that can be heat-labile (such as trypsin inhibitors and lectins) or heat-stable (e.g., saponins, phytates, condensed tannins, and oligosaccharides such as raffinose and stachyose); these factors negatively impact nutrient digestion and absorption [27,78]. The presence of these factors increases the risk of malnutrition and indigestion. Hence, it is desirable to reduce the content of these anti-nutritional factors via pretreatment before consumption [79]. Thermal treatment at 170–180 °C (e.g., by cooking or extrusion) can lower the levels of heat-labile anti-nutritional factors. However, treatment at such high temperatures can also damage heat-labile nutrients such as vitamins [80,81,82,83,84]. Enzymatic hydrolysis can also effectively decrease the amount of heat-stable anti-nutritional factors (e.g., phytic acid) present. However, excess enzyme activity can lead to macromolecular degradation, compromising the texture and integrity of the meat analog [85,86,87,88].

### 2.2. Protein Allergenicity and Ultra-Processed Food

Soy, wheat, and their derivatives have been identified as allergenic ingredients that can trigger frequent and severe reactions in some individuals [89]. Levels of heat-labile allergens can be reduced by thermal treatment. However, allergen degradation is rarely complete, and in some circumstances, the ingredient’s allergenicity may even worsen with treatment [89].

Plant proteins are globular and not fibrillar. Thus, processes and additives are applied to create fibrous structures from plant proteins. Employing industrial processes such as extruding, as well as molding, hydrolysis, hydrogenation, and reshaping or using chemical additives makes food treated this way an ultra-processed food according to the NOVA classification [90]. Generally, plant-based meat alternatives are considered ultra-processed products, and ultra-processed products have been linked to an increased incidence of obesity and cancer among consumers [91,92]. Furthermore, additives including texturizing agents (e.g., transglutaminases that increase intestinal permeability), as well as artificial colorants, and flavorings have been associated with celiac diseases and may be carcinogenic [93,94]. Due to these issues, there is a need to find safer alternative strategies for producing plant-based meat alternatives [15,95,96,97].

### 2.3. Digestibility and Nutrient Deficiency

Plant-based meat analogs show lower digestibility compared to real meat. Plant proteins often have a lower digestibility score (0.4–0.9) than animal proteins (more than 0.9) [72,73]. The bioaccessibility and bioavailability of plant ingredients, including starch, protein, and lipids, are crucial factors during digestion, therefore modulation of the plant nutrient microstructure might be essential [98]. Zhou et al. used an in vitro digestion standardized model to examine the digestive characteristics of a plant-based meat analog made from textured soy protein concentrate [99]. They concluded that two features were largely responsible for the poor digestion of meat analogs: a larger particle size than that of real meat, as well as the excessive usage of adhesive additives (mainly used to maintain the shape of the plant-based meat analog). Both features made it difficult for the digestive enzymes to contact the food ingredients, resulting in a lower digestibility. Furthermore, differences in the impacts of plant-based meat analogs and real meat on gastrointestinal functions in mice were demonstrated [100]. Mice (*n* = 16) were fed either plant-based meat analog or real meat diets for 68 days. Both diets included the same amount of protein content and numerous other nutrients, with the exception that the former contained more sodium, fiber, and fat, which cannot be entirely balanced based on the protein content. The study found that plant-based meat analogs had lower pepsin levels, associated with the generation of less peptides after digestion than meat. They also lowered the number of gastric parietal cells, as well as the levels of intracellular Ca^2+^, CAMK II, PKC, and PKA, as well as extracellular gastrin/CCKBR and Ach/AchR, all of which decreased gastric acid secretion capabilities. In the small intestine of plant-based meat analogs, there was a decrease in duodenal villus height and the ratio of villus height to crypt depth. Serum samples revealed lower levels of total amino acids, essential amino acids, and non-essential amino acids in plant-based meat analog groups compared to meat, indicating that plant-based meat analogs have poor in vivo absorption. The authors hypothesized that because of protein denaturation and aggregation during heating in the presence of salt and phosphate, a complex and rigid structure would form, resulting in a loss of protein digestibility. Furthermore, the authors identified differences in gut microbiota between the two diet types, which may also be connected with alterations in digestibility and absorption, and they urged further research [100].

Many plant proteins, such as legumes, also contain suboptimal levels of essential amino acids, particularly the sulfur-containing amino acids, including cysteine and methionine [101]. A leading role is played by methionine in a number of cellular functions, including the initiation of the translation of mRNA [102,103]. Besides being an essential structural and functional component of proteins and enzymes, cysteine is also required by other cell components containing reduced sulfur, such as methionine, homoglutathione, glutathione, iron–sulfur clusters, and vitamin cofactors such as thiamin and biotin, as well as multiple secondary metabolites [102,103,104]. Therefore, these two sulfur-containing amino acids are essential for dietary intake by humans. Furthermore, a metabolomic study using gas chromatography-mass spectrometry (GC-MS) analysis with electron impact ionization (EI) found essential metabolites including creatinine (product of creatine), hydroxyproline, anserine, glucosamine, and cysteamine (an aminothiol) to be present in real meat but not in plant-based meat analogs [105]. These nutrients have significant physiological, immunomodulatory, and anti-inflammatory roles in the human body, and their absence in the human diet has been associated with cardiovascular, retinal, neurological, and hepatic dysfunction [106,107]. Thus, fortifying plant-based meat alternatives with essential amino acids that include methionine and cysteine, as well as blending with different cereal proteins to avoid essential nutrients deficiencies, has been widely applied [108,109,110]. Iron from plant foods tends to be less bioavailable than iron from real meat [111]. Fortification remains the strategy of choice to overcome this decrease in iron bioavailability. However, its high cost and the presence of anti-nutritional factors in dietary components that inhibit iron absorption (e.g., phytates) make fortification less effective [112]. Furthermore, most of the plant-based meat alternative products available in the market contain less levels of vitamin B12 and zinc compared to real meat products [113,114,115,116,117]. Additionally, plant ingredients also tend to lack essential omega-3 fatty acids, especially the more bioavailable omega-3 fatty acid forms like eicosapentaenoic (EPA) and docosahexaenoic (DHA) acids. These fatty acids play vital roles in health maintenance, such as for cardiovascular, neurologic, and immune health [118,119]. To counter such deficiencies, some plant-based meat products are supplemented with omega-3 fatty acids. In summary, the currently reported nutrient deficiencies in plant ingredients constrain their application in meat analogs and manufacturers circumvent these issues by applying additives, which may achieve the target but increase the overall cost, and sometimes the health risks of consuming the product.

### 2.4. Food Spoilage and Pathogens

Food spoilage and the pathogens present in meat alternatives are considered health hazards. Plant-based meat analogs contain high protein and moisture levels and have a neutral pH value. These properties increase the proliferation of spoilage microorganisms and food pathogens in meat analogs [120,121]. A microbial survey by Tóth et al. to monitor the microbial quality of plant-based meat alternatives found the microbial load to be low during production [89]. However, large quantities of *Enterobacteriaceae* and yeast species were observed during storage in both refrigerated and unrefrigerated meat analogs. These contaminations likely originate from raw materials or during post-processing. These microbial species grew during storage, mainly at ambient temperature. As such, Tóth et al. concluded that uncooked plant-based meat analogs have a higher food safety risk than animal-based foods and additional precautions should be applied in their manufacture and storage. Foodborne pathogens may also be present in the plant ingredients, but these pathogens are mostly inactivated by exposure to heat during production (e.g., the extrusion process). However, spore-forming bacteria, including *Bacillus* spp. and *Clostridium* spp., may survive the heating process or contaminate the products after processing [79,122,123].

### 2.5. Genetically Modified Foods

There has been concern about applying recombinant proteins in the alternative meat industry. The recombinant proteins are foreign proteins created in prokaryotic and eukaryotic expression hosts [90]. Advances have been reported in using recombinant proteins to mimic the flavor and color of real meat in plant-based meat analogs [96]. However, the use of recombinant proteins still raises concerns due to their potential health and environmental risks [124]. Toxins, allergies, or genetic hazards are the main concerns regarding health risks linked with genetically modified food [124]. For instance, bean plants that were genetically altered to contain more cysteine and methionine were renounced when it was discovered that the transgene produced extremely allergenic proteins [125]. Similarly, altered metabolic pathways may result in the creation of toxins and other unidentified substances [125,126]. Table 2 shows a summary of the current major ingredients and techniques used in plant-based meat analog production, their positive contributions, and limitations in safety and nutrition of the final product.

## 3. Fermentation and Plant-Based Meat Analogs’ Nutrition and Safety

Meat alternatives should match real meat products in terms of nutrition as well as texture, flavor, and color to be widely accepted by consumers [128]. Recent studies highlight the public concern about the multiplicity of processing steps and utilization of additives in producing plant-based meat. Such approaches negatively impact the sustainability, safety, and nutrition of the final product [109,129]. Thus, it is necessary to optimize the quality of raw materials to reduce the number of additives and processing steps needed in manufacture while maximizing the products’ nutritional content. Mayer Labba et al. conducted a recent survey in Sweden to determine the nutritional content of 44 meat substitutes available in the market [130]. They discovered plant-based meat analogs to generally have a low iron content and high levels of phytate, highlighting the need to further improve the nutritional quality of existing meat substitutes. In contrast, tempeh and mycoprotein-based meat analogs (fermentation-based products) were found to be lower in phytates and higher in bioavailable zinc [130]. These results support our hypothesis that using fermented ingredients instead of raw or mechanically processed plant ingredients can improve the nutritional value of plant-based meat analogs. In the next subsection, we discuss how fermentation can alleviate the increasing health concerns associated with consuming plant-based meat analogs.

### 3.1. Anti-Nutrients

Studies have shown that fermentation can be used to partially or fully degrade anti-nutritional factors in food [131,132,133]. During fermentation, identified microorganisms were shown to impact the levels of anti-nutritional factors. *Bacillus subtilis* was shown to remove indigestible oligosaccharides from soybeans and improve soybean digestibility [134,135,136]. Similarly, the concentration of trypsin inhibitors, phytates, tannins, and oligosaccharides in fava beans and black beans significantly decreased after lactic acid bacteria (LAB) fermentation with *Weissella* spp. and *Leuconostoc* spp., *L. plantarum* and *L. casei* [137,138,139,140]. Yeasts such as *Kluyveromyces marxianus* [141] and *Lindnera saturnus* [142] were shown to reduce levels of anti-nutritional factors such as phytic acid, and trypsin inhibitors in soybean residues, in addition to removing the undesirable beany flavor. The filamentous fungus *Aspergillus oryzae* was isolated from soybean fermentation and reported to decrease the levels of trypsin inhibitors and phytic acids [16,143]. Furthermore, *Rhizopus* spp. and *Neurospora crassa* decreased the levels of glycinin, β-conglycinin, trypsin inhibitors, and oligosaccharides on food substrates [144,145]. Other legumes such as chickpeas and cowpeas fermented with *R*. *oligosporus* showed lower levels of oligosaccharides, tannins, and phytates in the fermented products [146,147]. Similar findings were achieved using coculture techniques. For example, *L*. *plantarum* and *L*. *acidophilus* were inoculated into cowpea and showed more effectiveness in degrading trypsin inhibitors than fermentation with individual inocula [148]. A mixed culture of *Bifidobacterium infantis* and *Streptococcus thermophilus* applied to soybeans significantly reduced the concentration of saponin and phytic acid upon fermentation [111,148]. These reductions were correlated with the microbes’ increasing enzymatic activities (e.g., proteases, phytases, phenolic oxidases, and glutathione reductases). For example, phytic acid complexes with numerous essential minerals, including zinc, calcium, and iron, making them unavailable for absorption in the body. Microbial enzymes, such as phytase, were found to degrade these complexes and unfold the bonding between the mineral and phosphorus in the phytate, making the minerals accessible in the body [149,150]. Similarly, proteinaceous antinutritional factors such as lectins that contain disulfide bonds are susceptible to microbial proteases such as glutathione reductase [151]. Lectin degradation during legume fermentation depends on the catalyzation of glutathione through thiol exchange reactions. Microbial activities such as organic acid creation and pH lowering by LAB can also promote anti-nutritional factor breakdown. This acidification process may support the natural endogenous enzymes in the beans that degrade these anti-nutritional factors [137,150,151,152]. It is worth mentioning that fermentation has also been shown to improve product stability by inhibiting the growth of spoilage microbes and food pathogens [153]. As an example, exopolysaccharides and organic acids synthesized during fermentation help to resist the growth of undesirable microbes and food pathogens. Such compounds also negate the effects of bacterial toxins, increasing product stability and reducing food poisoning [154,155].

### 3.2. Allergenicity

Fermentation may also reduce allergenicity of foods. Microbial isolates such as *Lactobacillus helveticus*, *L*. *casei*, *Enterococcus feacalis*, *B*. *subtilis*, and *A. oryzae* can degrade soy and gluten proteins into low molecular weight polypeptides, reducing their allergenicity [89,156,157,158,159]. Moreover, studies revealed that structurally altering allergen protein conformations, including increasing surface hydrophobicity and β-strands, decreased the allergenicity of soy protein by roughly 90% following fermentation by *L*. *plantarum* [160]. This conformation might be related to a decrease in pH, which leads to the disruption of the soy protein structures and a loss of allergenicity. Sun et al. reported a significant reduction in β-glycinin lgE reactivity after fermentation by *L*. *plantarum*, as a result of lactic acid production during fermentation [161]. Particle size distribution analysis showed that fermentation induced the formation of protein gel/aggregates into large particles at pH 4.5 and reduced band intensities of α-, α′-, and β-subunits from SDS-PAGE analysis, suggesting a transformation of soluble to insoluble proteins. The β-subunit of β-glycinin was reported to play an important role in immunoreactivity, as it contains many lgE-binding epitopes [161]. Disruption of the protein subunits and aggregation of the protein units during fermentation may have led to the burial of these epitopes located on the surface of the β-subunit, resulting in a reduction in the lgE binding capacity and a reduction in immunoreactivity [162]. However, the pH at the end point of fermentation also plays a crucial role in immunoreactivity reduction. At pH 4 and pH 3.5, an increase in band intensities of the proteins was reported, suggesting a dissociation of the protein into smaller particles and a disruption of the gel/aggregation matrix. This degradation of the matrix due to a looser protein structure may have led to a release in the subunits and expositing lgE binding epitopes, thereby increasing immunoreactivity [162]. Frias et al. also compared the rate of immunoreactivity degradation in soybean flour and reported lactic acid bacteria to result in a higher degradation rate as compared to fungi when used as starter cultures in fermentation [163]. Peptide analysis revealed smaller particle size and the presence of less intense immunoreactive peptides below 30 kDA, with reductions greater than 90% in fermentation using *L. plantarum* as a starter culture [163]. Therefore, protein degradation and conformation changes were responsible for the reduction of fermented protein allergenicity.

### 3.3. Digestability

Potential improvements in digestibility were observed in plant proteins fermented with *A. niger* [164], *Ligilactobacillus salivarius* [165], and naturally [166,167]. As an example, soybean fermented with *N. crassa* led to 10.5% protein hydrolysis and a 13-fold increase in the levels of free amino acids that was associated with a 37.9% rise in the in vitro protein digestibility [168]. Similarly, fermentation can break up complex polysaccharides and fat present in plants and produce lower molecular weight compounds, as discussed above. As an example, the in vitro starch digestibility of unfermented black gram (legume) was found to be 35.7 mg maltose released/g [169]. Spontaneous fermentation improved the starch digestibility to 59% in 18 h at 25 °C, which was further improved to 88% when fermentation was conducted at 35 °C. This breakdown that occurred during fermentation was thought to result from the activities of endogenous enzymes from either the endogenous microbiota or the legume [169,170].

### 3.4. Improve Nutritional Components

Fermentation also improved the amino acid profiles of the plant-based substrates used in meat alternatives [153]. For instance, *L. plantarum*, *L. acidophilus,* and *Bifidobacterium* species were shown to increase the total protein concentration and the levels of methionine, tryptophan, and lysine in fermented soy-based products [171,172,173]. This increase in the total protein content and the changes in the amino acid profile observed in the fermented sample may be correlated to the microbial inoculum itself, its growth, or its enzymatic activities [174]. Microorganisms use carbohydrates as a carbon source and convert them into microbial proteins through intermediary metabolism [141]. They also break down the feed protein into short peptides and free amino acids, as mentioned above. These activities illustrate the potential changes that may occur to the amino acid profile during fermentation. Fermentation can increase the levels of bioactive compounds that are beneficial to human health. High levels of phenolic, flavonoid, antioxidant, and antimicrobial compounds were detected in soybeans fermented with *N*. *crassa* [168], *Monascus purpureus*, *A. oryzae* [175], *B. subtilis* [176], *B. velezensis*, and *Pichia anomala* [177]. Microorganisms produce secondary metabolites during growth, including alkaloids, terpenoids, phenols, steroids, peptides, flavonoids, polyketones, and quinols. These substances have vital functions in microbial growth, such as adaptability, defense, and signaling during environmental stresses or ecological interactions [178]. They are biosynthesized in microorganisms largely by the shikimate and phenylpropanoid metabolic pathways [179]. These compounds have physiological functions if consumed in sufficient amounts. Alkaloids, for example, are biosynthesized from amino acids such as lysine, ornithine, aspartic acid, tyrosine, phenylalanine, and tryptophan and show anti-cancer effects [180]. Polyphenols have been reported to have antioxidant, anti-carcinogenic, and anti-microbial activities [181]. Similarly, some steroids, peptides, polyketones, and quinols were reported to have essential health benefits for humans [182]. Besides biosynthesis, microbial enzymes may also help to increase the bioavailability of some of these compounds, if they occur naturally in the plant matrix. For example, microbial enzymes including cellulase, amylase, xylanase, esterase, and β-glucosidase have been reported to catalyze covalent bond hydrolysis between lignocellulose and phenolics in the plant matrix, releasing phenolic compounds. These results were observed during the fermentation of mulberry fruits and leaves by microorganisms, including *L. plantarum*, *Saccharomyces cerevisiae*, *M. purpureus*, and *R. oligosporus* in studies [183,184,185]. Similar results were reported for the biovailability of other nutrients. As an example, the fermentation of soy curd using a mixture of the yeast *S. boulardii* and LAB *L. plantarum*, was associated with significant increases in calcium and magnesium bioavailability [148,186]. These results are associated with a reduction of the antinutrients, mainly phytic acid, which correlate to increased mineral availability. Tangyu et al. found the fermented product to contain lower levels of oligosaccharides (stachyose and raffinose) and beany flavors [148]. Surprisingly, the degradation of phytates by phytase that can occur during fermentation has been shown to produce metabolically active myo-inositol phosphates with potential health benefits [187], with D-myo-inositol (1,3,4,5) tetrakisphosphate and D-myo-inositol (1,4,5) trisphosphate reported to have anti-inflammatory and anti-tumor activities, as well as preventing diabetes complication and promoting heart health [188,189]. As mentioned, many plant ingredients do not contain all the essential amino acids or the omega-3 fatty acid DHA, necessitating the addition of these ingredients. Using a specific microbial group, such as microalgae, may provide an alternative method for plant-based meat substitutes. Microalgae proteins contain all the essential amino acids and some microalgae, such as *Schizochytrium* spp., are considered a valuable source of the omega-3 fatty acid DHA [111,190]. Additionally, microalgae such as *Arthrospira platensis* have been reported to contain more calcium than raw milk (about 180% of raw milk), making it a good source of calcium [191]. Xia et al. mixed the *Haematococcus pluvialis* residue with traditional plant pea protein and found a significant improvement in the final product appearance and texture to mimic the real meat [192]. Besides using microalgae as ingredients, they can also be employed as a fermenting group. Several studies showed the capability of various microalgae strains to be cultivated on different agricultural processed residues including soybeans, pea seed, and corn seed [193,194,195,196]. Additionally, the development of consortia of microalgae with bacterial or fungal species has recently widened the microalgae applications [197]. Such characteristics might make microalgae a suitable candidate either to be used as an additive in plant-based meat analogs or combined with other microorganisms (consortia) during fermenting plant-based raw ingredients.

### 3.5. Others

Finally, applying probiotics to plant-based meat analogs can confer other health benefits to plant-based meat analogs if minimal heat treatment is applied or selected thermotolerant strains are used to ensure probiotic livability. Fermentation using probiotic species, such as *L. acidophilus*, *L. delbrueckii*, *L. salivarius*, *Clostridium butyricum*, and *S. boulardii* were reported to confer health benefits related to probiotic consumption, including enhanced intestinal health, improved immune response, reduced cholesterol level, and cancer prevention [160,161,162]. Plant proteins have been shown to be efficient probiotic carriers [198]. Probiotic LAB species including *L. casei*, *L. fermentum*, *L. helveticus*, *L. reuteri*, *L. rhamnosus, L. acidophilus*, *L. johnsonii*, *B. animalis* ssp. *lactis*, *E. faecium*, and *S. thermophilus* have been successfully applied in plant-based beverages, cheese alternatives, and yogurt alternatives [199]. Soy protein isolate (SPI) was used to carry *L. paracasei* and showed substantial protection in simulated gastrointestinal conditions (45 g/L bile salt, 8.5 g/L NaCl, and 1 g/L pancreatin at pH 8) [200]. Here, Yan et al. combined a bacterial solution with an interpenetrating polymer network made of soy protein isolates and sugar beet pectin (SBP). Different concentrations of laccase were added to the mixture, which enhanced the formation of a hydrogel by creating a crosslinked network between the protein and the pectin. This study demonstrated that the produced gel preserved over 96.7% of the bacterial viability when treated to stimulated gastric fluid, with no viable cells observed in the free culture. Moreover, *L. paracasei* showed significant stability during storage at 4 °C for 21 days, with maximum stability found at 3.5% SBP, 10% SPI, and 10 U laccase, which correlates to a higher water holding capacity of the formed gel [200]. The presence of protein and pectin in plant ingredients, together with various microbial enzymes created during fermentation, can make plant-based meat analogs an excellent delivery system for probiotics and boost their nutritional quality. These features make the use of microorganisms and fermentation techniques promising avenues for improving plant-based meat alternative safety by reducing anti-nutritional factor levels, increasing digestibility, providing probiotic health benefits, and including high-value nutrients such as polyphenols, vitamins, minerals, essential amino acids, and omega-3 fatty acids.

### 3.6. Nutrition and Health Challenges

While fermented food is often considered a ‘super food’ that is rich in bioactive compounds including vitamins, peptides, minerals, and organic acids, they may also be associated with negative health impacts. Toxic metabolites such as biogenic amines (BA) and mycotoxins have been reported with fermenting microorganisms that include bacteria and fungi. Mycotoxins including aflatoxins, ochratoxins, and patulin have been reported to be synthesized by many species of *Aspergillus* and *Penicillium*, while trichothecenes, fumonisins, and zearalenone compounds may be produced by *Fusarium* species [201]. Some of these mycotoxins are known carcinogens, for instance, the ones synthesized by *A. parasiticus* and *A. flavus*. Others produced by *Fusarium* spp. may increase the host’s susceptibility to infectious diseases [201]. Several treatments have been proven to reduce the level of mycotoxigenic fungi and mycotoxins before fermentation. Sorting, soaking, washing, and cooking, as well as the addition of organic acids, bases, or oxidants during the soaking, have all been shown to decrease their presence in soybeans [202,203]. Furthermore, adjusting the fermentation conditions may directly reduce the presence of mycotoxigenic fungi and mycotoxin generation. More specifically, optimizing the temperature, pH, and water activity of the fermentation mass to levels that favor the growth of the inoculum may enable them to lead the fermentation process and prevent the presence of undesirable microorganisms, including filamentous fungi [203]. For the inoculum criteria, selecting non-mycotoxin producing strains is essential to avoid the mycotoxin generation. Additionally, various microorganisms have been reported to suppress the growth of mycotoxin-producing species or to degrade mycotoxins. For example, *B*. *licheniformis* was shown to outcompete mycotoxigenic *Aspergillus* spp. [204], while *B*. *albus* was capable to metabolize aflatoxin B1 and G1 and significantly decrease their concentrations [205]. Both capabilities, the growth suppression of undesirable fungi and mycotoxin degradation, were also observed with *A*. *oryzae* strains. These strains were isolated from fermented soybean products and demonstrated the ability to suppress the growth of mycotoxigenic *A*. *flavus*, as well as breakdown more than 90% of aflatoxin B1 present in culture broth [206]. 

Biogenic amines, such as histamine, tyramine, cadaverine, and putrescine, are low molecular weight nitrogenous compounds produced by the decarboxylation of amino acids including histidine, ornithine, and tyrosine [207]. They have been detected in microbial isolates that include genera belonging to Enterobacteriaceae [208,209]. Heat treatment, such as pasteurization, as well as the addition of protectors and oxygen scavengers are the most common strategies to suppress the growth of Enterobacteriaceae isolates [210]. Some *Bacillus* strains such as *B. subtilis* and *B. amyloliquefaciens* have been reported to produce putrescine and cadaverine during sausage fermentation [211]. Similarly, LAB fermentation with the genera *Lactobacillus*, *Enterococcus*, *Lactococcus*, *Leuconostoc*, and *Streptococcus* can also produce BA in fermented dairy products [210,212,213]. Different measures have been applied to prevent or decrease the BA during sausages and dairy fermentation including heat treatment, improvements to the hygiene level during production, and preservatives [211,214,215]. Additionally, fermentation of foods and beverages with non-BA producing microorganisms or microorganisms that degrade BA are essential approaches to reduce BA levels. For example, inoculating soybeans with *B*. *subtilis* T2 resulted in over 80% reduction in total BA content compared to spontaneous fermentation with the endogenous species [216]. For the BA degradation, a number of microbial enzymes, including monoamine oxidases, diamine oxidases, and multicopper oxidases, have been reported to degrade the BA [217]. Such enzymes were correlated to metabolize tyramine and putrescine by *L*. casei and *L*. *plantarum* [218,219,220], as well as to degrade histamine by *Debaryomyces hansenii*, during evaluation studies [221]. Consequently, choosing the right microorganism and formulating strategies to improve safety is essential for increasing the reliability and applicability of fermentation in plant-based meat analogs. A selection of nutritional and safety characteristics of selected microorganisms is summarized in Table 3.

## 4. Fermentation and Plant-Based Meat Analogs’ Sensory Quality

The main focus of this review is on plant-based meat analogs safety and nutritional quality. However, in this part, we aim to shed light on the impact of fermentation on sensory quality as a key characteristic of the meat analogs. As mentioned above, plant-based meat analogs are largely influenced by their formulation and structuring technology. The sensory characteristics of plant-based meat analogs have been reviewed in depth by other authors [7,39,40,41,42,121]. Briefly, several key ingredients are used in the production of plant-based meat analogs. These ingredients contribute to the color, flavor, and texture of the product. The sensory quality of plant-based meat analogs can be improved, but there are several drawbacks. Plant ingredients lack a perceptible meaty aroma and taste, as well as having off flavors such as green, beany, astringent, bitter, and metallic tastes [224,225]. There has been limited success reported with flavorings to mask off flavors or reproduce meaty flavors, and some flavoring agents may be destroyed during cooking [226]. In terms of texture, extrusion is the most common method used. Nevertheless, extrusion presents challenges in producing cost-effective fibrous meat-like structures. In addition, several parameters must be closely monitored, including barrel temperature, pressure, and powder/water feed rates [40,227,228]. A similar constraint is color, since plant proteins typically lack the red or brown color associated with raw or cooked meat. Heat-stable coloring compounds such as annatto, caramel, and carotene are used to simulate the red color of raw meat; however, they cannot replicate the color of cooked meat. In general, existing techniques have had little success in reproducing the real meaty sensory quality [129,229,230,231]. The use of microbial fermentation may provide an alternative method of replicating these characteristics.

The beany flavor of plant materials can be successfully minimized or eliminated through the fermentation process. Researchers have demonstrated that soybean off-flavor characteristics, such as the beany flavor, decreased or disappeared after fermentation using different microorganisms, including *Kluyveromyces marxianus*, *B. subtilis*, and *Weissella confuse* [134,141,232]. It is believed that these microorganisms are capable of degrading lipoxygenases, which act on polyunsaturated fatty acids to produce off tastes such as beany notes. Additionally, by generating pleasant microbial volatiles, they can also mask these undesirable odors. In a similar manner, undesirable aldehydes can be transformed into desired chemicals by microbes, such as *Lindnera saturnus*, through the metabolism of ester compounds during fermentation [142]. An increase of 70 times in the concentration of esters, ranging from 0.17 to 0.28 mg/g dry weight, as well as other important volatiles, including ethyl heptanoate, hexyl acetate, 3-hexenyl acetate, octanoate, and 2-heptenyl acetates, were detected. These findings demonstrate that microorganisms have the capacity to reduce off-flavors by either denaturing the enzymes directly, degrading undesirable aldehyde compounds or masking them through the generation of desirable metabolites.

Despite the fact that plant-based meat components are primarily composed of proteins, polysaccharides, and lipids, they lack important intermediary substances like reducing sugars, fatty acids, and amino acids that are necessary to produce distinct meat aromas and flavors [233,234]. Many bacteria were found to have strong enzymatic activity, which may aid in the breakdown of these complex molecules. For instance, several microorganisms, such as *B*. *subtilis*, *B. polyfermenticus*, and *B. amyloliquefaciens*, demonstrated high protease activity and produced significant quantities of peptides and amino acids in fermented products [134,135,235,236,237]. At 35 °C and pH 7, *R*. *oryzae* and *Mucor* sp. have been shown to produce a potent lipase activity that breaks down lipids into small peptides and fatty acids [238,239]. The presence of these degraded compounds may increase the formation of desired flavors and produce flavors during further processing (heating) [233,240]. 

As mentioned above, current texturing techniques have some difficulties in creating the mouth feeling, a fibrous structure, and a meaty appearance. Microorganisms and fermentation techniques can be used to overcome these limitations. For example, a *B. subtilis* fermentation step during the production of a meat analogue results in a product with desirable eating qualities, improved chewiness, integrity, and firmness, when compared to a non-fermented product [241]. Similarly, the functional properties of various fermented products are enhanced by filamentous strains such as *A*. *oryzae*, *R*. *oryzae*, *Fusarium venenatum,* and *Neurospora intermedia* [242,243,244,245,246,247]. Such filamentous species have high-quality proteins, and their mycelia are rich in fiber and polyunsaturated fatty acids, which are modified through controlled denaturation processes to give them a meaty texture [17,248].

Fermentation is shown to improve the organoleptic quality of the final products. However, it is worth noting that excessive microbial activity during fermentation, mainly enzymes and generating organic acids and microbial volatiles, can be associated with unfavorable changes in product sensation and texture. For example, over-fermentation can lead to the accumulation of flavor compounds such as esters, creating an overly fruity aroma. It can also cause the production of other fermentation off flavors such as propionic acid, ferulic acid, and 2,3-pentanedione, which can adversely affect the organoleptic quality of fermented products [249]. Similarly, microbial enzymes and frequent acidification processes during fermentation may be associated with significant changes in product texture, and their suitability for plant-based meat alternatives should be further investigated. For example, cassava starch fermented with *B*. *subtills*, *L*. *plantarum,* and *Candida krusei* showed lower water absorption, lower swelling capacity, lower adhesiveness, and lower viscosity compared to non-fermented samples [250]. These findings highlight the importance of considering and monitoring sensory effects in order to effectively utilize fermentation to improve the safety and nutritional features of plant-based meat analogs.

## 5. Plant-Based Meat Analogs and Starter Culture Technology

Fermentation technology may be a good way to improve the quality and acceptability of plant-based meat and starter cultures can be used to modulate fermentation outcomes. So far, no starter cultures have been designed to fulfill these characteristics. The success stories of developing new starter cultures for numerous foods gives us optimism about its eventual adoption here. Starter cultures have been used to improve the texture, flavor, appearance, and nutritional quality of tempeh, bread, cheese, yogurt, coffee, and sausage to satisfy consumer preferences [37,78,251,252,253,254,255]. Given the crucial role of microorganisms in improving the characteristics of plant ingredients, it is clear that starter cultures show great potential in producing ingredients that better mimic the characteristics of real meat and overcome the current disadvantages associated with the use of additives or extensive processing steps. To choose the best starter cultures, it is important to assess the capacity of the microorganisms to carry out the desired biotransformation and their potential for commercial development (Figure 1). Laboratory assays have been used in the traditional microbial screening process for these characteristics, and modern bioinformatics tools such as the Kyoto Encyclopedia of Genes and Genomes (KEGG) database may also be used to help scientists look for suitable microbial candidates [256,257,258]. The general recommendations for designing a starter culture to enhance safety and nutrition in plant-based meat analogs are given below.

Based on the above discussion, most ingredients used for producing plant-based meat, such as legumes, have significantly high protein, carbohydrate, and fat concentrations, as well as anti-nutritional factors. Starter cultures with strong enzymatic activity, including proteases, lipases, amylases, and phytases, are needed to transform these components. Degradation of these substances improves the digestibility of the final product, as well as reducing the allergenicity and anti-nutritional factor content of plant-based meat, as mentioned above. Furthermore, attention should be given to microorganisms known for producing desired volatile and non-volatile profiles. Conventional and advanced methodologies that have been applied to measure these microbial activities in other areas of study include plate assay, colorimetry, chromatography, microcalorimetry, and sensory tests, all of which can be applied to microbial screening [259,260,261].The fermentation of the plant ingredients used in meat analogs has been linked to health benefits including an increase in the level of essential amino acids, omega-3 fatty acids, bioactive compounds, probiotics, and an improvement in the meat analog’s safety and stability. These features have potential physiological roles in the human body and should be considered when screening for suitable starter cultures. Selecting the right microorganisms with such characteristics as the main fermenting microorganism or as the coculture may boost the acceptability of plant-based meat analogs. Laboratory and clinical studies that have been widely applied to test the safety and health benefits of fermented products may be employed in the microbial screening process for plant-based meat analogs [258,262,263].Using the available microbial survey data of plant ingredients and meat analogs, the selected strains should be examined for their ability to adapt, compete with the natural microflora, as well as other microbial contaminates and food pathogens, that may present in the raw ingredients during and after processing. An in-depth investigation of microbial safety, and biodegradation capability to toxic compounds include mycotoxins and biogenic amines, should be considered. This criterion can be determined by exposing the selected strains to different stressors (such as high temperature, high salt, pH, and other additives), as well as observing how they react to microflora, foodborne pathogens, and toxic compounds that are frequently found in raw ingredients and processed foods. Successful growth under such stressful conditions is considered a potential indicator of high fermentation performance of the selected isolates. Additionally, factors such as inoculum size, inoculation time, and incubation parameters should be controlled to ensure successful fermentation with desirable results. Similar approaches have been applied to develop starter cultures for other food products [264,265,266].Additionally, for commercial applications, selected strains designed for starter culture should be able to be cultivated on available and cheap substrates to lower production costs. In addition, the strains should tolerate downstream processes such as air drying, freeze drying, packaging, and rehydration to ensure stability during storage and handling [267,268].With current advancements in molecular techniques, screening and gene editing may be used to increase the capability of the selected isolates to desirably interact with the food matrix. A similar approach was used to improve LAB strains in the meat and dairy fermentation process, which involved no extra risk compared to the use of wild strains [269,270,271]. Genome editing technologies, like CRISPR-Cas9, can be used to eliminate specific DNA sequences from a microbial genome that control mycotoxin or BA biosynthesis, or to add desired genes that biocontrol undesired microorganisms and toxins. Despite the fact that these applications can reduce costs, and improve strain capabilities, using genetically modified organisms (GMOs) in food may trigger public concerns [206,272,273].Before starting the development of starter cultures and their commercialization, the Nagoya protocol should be considered. Based on this protocol, prior informed consent and mutually agreed terms must be built by the research provider describing access to the resources and benefit shares [274].

## 6. Conclusions

Persuading the general population to adopt plant-based meat analogs as a protein source has been challenging due to difficulties in mimicking the taste and texture of real meat. Furthermore, the current approach of making plant-based meat alternatives by using different ingredients and heavy mechanical processes is costly and has health concerns. Our survey of the literature indicates fermentation with different microorganisms to be an efficient way of overcoming these drawbacks and reducing the need for excessive additives and heavy processes. Fermentation additionally reduces the anti-nutritive levels and allergenicity, as well as increases digestibility and micronutrient bioavailability, and improves safety and sensory quality. To support the market growth of plant-based meat alternatives, additional studies are required to advance our understanding of how fermentation and starter culture technology can improve the quality of plant-based meat alternatives, with special consideration to the final product’s sensory quality.

## Figures and Tables

**Figure 1 foods-12-03222-f001:**
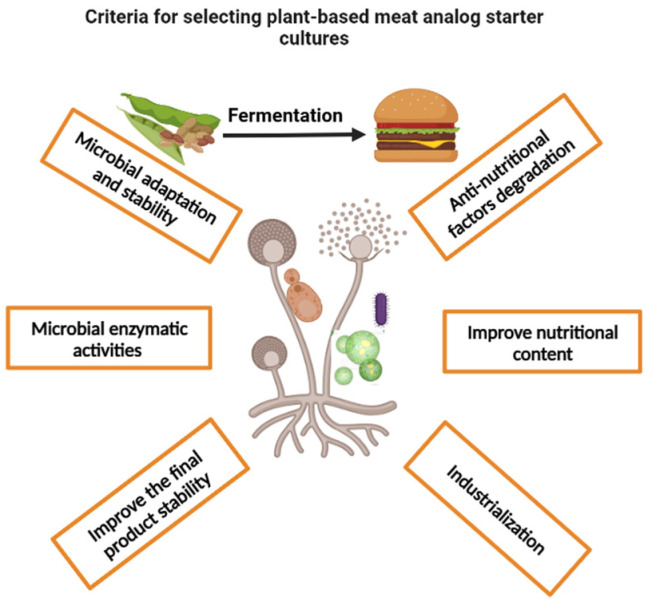
Starter cultures selecting criteria for plant-based meat analogs to improve the nutrient and safety aspects, created by BioRender.com.

**Table 1 foods-12-03222-t001:** Comparison of nutritional content, selected advantages and disadvantages of conventional meat and plant-based meat alternatives.

Food	Nutritional Content (%)			Advantages	Disadvantages	References
	Protein	Fat	PDCAAS ^1^			
Meat				High protein content and protein digestibility High consumer acceptance and sensory quality Good source of iron and vitamin B12	Resource-intensive production Animal-welfare concerns Red meat consumption linked to adverse health effects (e.g., cancer and cardiovascular disease)	
Chicken	22.3–22.7	0.9–2.1	0.95			[43,44,45]
Beef	20.6–22.5	4.3–6.8	0.92			[43,44,46]
Pork	21.8	4.0	-			[44,47]
Mutton	20.2–21.6	4.6–8.0	0.99			[43,44]
Meat alternatives						
Tempeh (fermented whole soybean)	61.9–56.9	8.4–23.9	0.92–0.99	Good source of protein, low in saturated fat Good source of iron and fiber Free of cholesterol High digestibility More resource-efficient production than meat Low allergenicity, fermentation breaks down allergenic proteins	Lack sulfur-containing amino acids, including methionine and cysteine Lack of vitamin B12, except if vitamin B12-producing bacteria present during fermentation Low consumer acceptance Sensory quality is different from meat	[48,49,50]
Tofu (made from soymilk)	11.3	7.84	0.56–0.70	Rich in B vitamins, and low in sodium Net protein utilization (NPU) is estimated to be around 65%, making it comparable to chicken meat in terms of assimilation and digestion More resource efficient production than meat	Lack sulfur-containing amino acids Lower digestibility than meat Loss of nutritional and nutraceutical contents during processing Presence of anti-nutritive factors Lack flavors causing a low consumer acceptability	[51,52,53,54]
Seitan (made from wheat gluten)	34.3	0.78	0. 23	Consumption of 100 g provides 61.2–74.5% of recommended daily protein	Low in lysine	[53,55,56]
				Its fibrous structure and high protein content make it an excellent meat substitute	Low digestibility	
				Its sensory properties can be easily modulated by spices and flavors during manufacturing due to its neutral taste and aroma	Sensory quality is closer to meat than tempeh and tofu but still not a perfect real meat analogy	
Quorn (mycoprotein, made from *Fusarium venenatum*)	9.4–11.5	2.6	0.91	High protein digestibility, low in saturated fat Low antinutrient content More resource-efficient production than meat High fiber content Texture more like meat compared to plant proteins	Lower levels of iron and vitamin B12 than real meat May cause allergies and/or gastrointestinal symptoms Possible presence of mycotoxins after inoculating *F. venenatum* into rice culture	[57,58,59,60,61]
Texturized plant protein ^2^ Soybean isolates Wheat gluten Pea protein concentrates	87.0 80.0 50.0–85.0	<0.1 - <1	~1.0 0.26 0.73	High protein content, low in saturated fat, free from cholesterol Fibrous structure and texture like meat Possible to blend protein sources to achieve a more complete amino acid profile	Deficient in micronutrients that are common in meat (e.g., vitamin B12 and iron) Considered as ultra-processed foods associated with adverse health effects Usually not clean label as additives are added to modulate the sensory properties (e.g., texture, color, and flavor). These additives may not diffuse in the product homogeneously, leading the worse sensory quality than meat	[62,63,64,65,66,67,68]

^1^ PDCAAS = Protein Digestibility Corrected Amino Acid Score; ^2^ most common protein sources, which are used in plant-based meat analogs.

**Table 2 foods-12-03222-t002:** Main approaches used in plant-based meat analog production to improve safety and nutrition of the final products, their positive contribution, and current limitations.

Target	Ingredients and Processes	Functions	Limitation of the Current Methods	References
Enhance product safety	Heat Add ascorbic acid, essential oils, curcumin, polyphenols, tocopherols, spices, carotenoids, and herbs	Minimize product contamination and food poisoning Improve product shelf-life and health	Survival of food spoilage and pathogens Resistance of anti-nutritional factors, such as saponins, alkaloids, phytates Failure to completely remove allergens such as soybean protein and gluten Some used additives are correlated to human diseases and public concerns Considered as ultra-processed products cause obesity and cancer	[70,77,89,91,92,112,122,127]
Improve product nutrition	Blend proteins, carbohydrates, and oils Fortification and encapsulation for micronutrients, including minerals, and vitamins	Qualify as good sources of protein, energy, and fiber Increases the concentration and bioavailability of essential nutrients overcome their deficiencies	The extensiveness of processes and functional ingredients and additives make it an expensive purchase Damage heat-labile nutrients during processing Presence of phytates reduces bioavailability of essential minerals	[74,75,76,112,119]

**Table 3 foods-12-03222-t003:** Microorganisms’ contributions to the nutrition and safety of plant-based meat alternatives.

Fermentation by	Contributions to	References
*Bacillus subtilis/Bacillus velezensis/* *Ligilactobacillus salivarius/Weissella* spp./*Leuconostoc* spp./*Lactiplantibacillus plantarum* *Lactobacillus casei/* *Pichia anomala/Saccharomyces cerevisiae/* *Neurospora crassa/Monascus purpureus/Aspergillus oryzae/* *Rhizopus oligosporus*	Improves digestibility (breakdown of polysaccharides, proteins, and lipids)	[134,135,136,164,165,166,167]
*Weissella* spp./*Leuconostoc* spp. *L. plantarum*/ *L*. *casei*	Decreases trypsin inhibitors, phytates, tannins, and convicine	[137,139,140]
*Kluyveromyces marxianus/* *Lindnera saturnus*	Decreases phytic acid and trypsin inhibitors	[141,142]
*A*. *oryzae*	Reduces trypsin inhibitors and phytic acid	[16,143]
*Rhizopus* spp./*N. crassa*	Reduces glycinin, β-conglycinin, trypsin inhibitors, and oligosaccharides	[144,145]
*L*. *casei/Lacticaseibacillus helveticus/* *Enterococcus faecalis/* *B. subtilis/A. oryzae*	Reduces allergenicity	[89,156,157,158,159]
*Bifidobacterium* species	Increases protein concentration	[171]
*L. plantarum/* *L*. *acidophilus*	Increases the levels of methionine, tryptophan, and lysine Competes and reduces the growth of spoilage and pathogenic microorganisms	[153,172,173]
*B*. *subtilis/B*. *velezensis/L*. *plantarum/* *P. anomala/S*. *cerevisiae/N. crassa/M. purpureus/A*. *oryzae/R*. *oligosporus*	Increases phenolics, flavonoids, antioxidants, and antimicrobials Enhances digestibility Decreases allergenicity	[168,175,176,177]
*L*. *acidophilus/L*. *delbrueckii/L*. *salivarius*/*C*. *butyricum/S*. *boulardii*	Probiotics health benefits	[222,223]
*Aspergillus* spp./*Penicillium* spp./*Fusarium* spp.	Secretes mycotoxins (carcinogens) Decreases immunity	[201]
*B*. *subtilis/B*. *amyloliquefaciens* *Lactobacillus* spp./*Enterococcus* spp./*Lactococcus* spp./*Leuconostoc* spp./*Streptococcus* spp.	Forms biogenic amines	[210,212,213]

## Data Availability

Data will be available upon request.

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
