# Peer review of "Significance of Fermentation in Plant-Based Meat Analogs: A Critical Review of Nutrition, and Safety-Related Aspects"

_foods, 2023, doi:10.3390/foods12173222_

Round 1
Reviewer 1 Report
The manuscript “Significance of fermentation in plant-based meat analogs: a critical review on nutrition, and safety-related aspects” by Elhalis et al. surveys the literature related to the current status of plant-based meat analogue developments and to the effects of fermentation on nutritional quality and safety of different types of (plant) raw materials.
Based on the title, the manuscript deals with a timely and important topic. Throughout the manuscript, the authors prove to be well aware of both the nutritional and technological benefits and challenges related with plant-based meat analogues and fermented foods. However, the manuscript lacks coherence and does not provide any truly new viewpoints on fermentation as a tool in the development and processing of meat analogues, per se. In addition, I have some concerns about the used references as the references include papers that have been supported by food industry operators (e.g., Fraser et al. [69], Wu, [74]), which in my opinion, are not suitable references for a critical review article. Quite high proportion of the references seem to be review articles and they are often cited in cases where the original research article should be referred to.
For the above-mentioned reasons, I suggest that this manuscript will be rejected.
Please see my recommendation above.
Author Response
We thank and understand the reviewer comments and concern. In this manuscript, we explore fermentation as a possible way to meet the current requirements for plant-based meat analogs, explaining its potential and the associated safety concerns, ends with a recommendation and future works.
Herby, we cannot find a better way to explain the paper aim and structure more than quoting reviewer 4 comment below.
''The authors' intention to bridge this knowledge gap is commendable. By meticulously analyzing the current understanding of nutrition and safety in plant-based meat alternatives, they lay the groundwork for significant improvements. The proposed employment of fermentation as a viable approach to address these limitations is a stroke of brilliance.
Notably, the article skillfully outlines the main methods employed to enhance the nutritional content and stability of plant-based meat analogs, including the use of plant protein blends, fortification, and preservatives. However, the authors acknowledge that recent concerns have arisen regarding safety, nutrient deficiencies, low digestibility, allergenicity, and high costs.
Here, the authors highlight the transformational role of fermentation with diverse microorganisms, such as Bacillus subtilis, Lactiplantibacillus plantarum, Neurospora intermedia, and Rhizopus oryzae. This process has been shown to significantly improve digestibility while effectively reducing allergenicity and antinutritive factors. Additionally, the incorporation of various microbial metabolites enhances the safety and nutrition of the final product, though some concerns remain.
The paper's proposal to design a single starter culture or microbial consortium for plant-based meat alternatives is a revolutionary concept. This approach not only enhances the health benefits of the end product but also aligns perfectly with the demands of an expanding and sustainable economy.
A key highlight of the article is its focus on persuading consumers to embrace plant-based meat analogs as a protein source. The difficulties in replicating the sensory experience of real meat have hindered this transition. Moreover, the current resource-intensive and mechanically driven manufacturing processes raise both cost and health-related issues. Here, the authors effectively argue that fermentation with diverse microorganisms offers an efficient solution to overcome these challenges. By reducing anti-nutritive levels and allergenicity while enhancing digestibility and micronutrient bioavailability, fermentation significantly enhances the appeal of plant-based meat alternatives.
The call for further research to explore the potential of fermentation and starter culture technology in improving the quality of plant-based meat analogs, with particular attention to sensory quality, is both timely and crucial. As the market for plant-based alternatives grows, advancing our understanding in this domain will undoubtedly pave the way for a more sustainable and enticing future.
In conclusion, the article "Significance of fermentation in plant-based meat analogs" deserves accolades for its pioneering approach in addressing critical aspects of plant-based meat alternatives. By spotlighting fermentation as a transformative solution, the authors have presented a roadmap towards safer, more nutritious, and appealing plant-based meat products. This review serves as an invaluable resource for researchers, policymakers, and industries committed to shaping a healthier and more environmentally conscious future.
C. the used references as the references include papers that have been supported by food industry operators (e.g., Fraser et al. [69], Wu, [74]), which in my opinion, are not suitable references for a critical review article.
R: All the references in this manuscript, including those referred to, has been used in published papers. We believe that including these types of references is essential, because we are discussing current limitations of a commercial product and how science and technology can help to improve it and keep it forward.
C. Quite high proportion of the references seem to be review articles and they are often cited in cases where the original research article should be referred to.
R. We use original research articles, mainly when we feel the readers may need more related information which out of our focus, as well as, when we come to a conclusion and gap of knowledge, as a support.
Thank you.
Reviewer 2 Report
The manuscript is a review about fermentation in plant based meat analogs.
The manuscript is well written and presented.
However, the authors should add a table including a comparaison between nutrionnal aspect (proteins, fat, aminos acids, etc.) of meat and fermented/ meat analogs with negatives and positives points of each food.
Author Response
We thank the reviewer for the crucial suggestion of the table. Attached the table illustrated key related findings of compression of nutritional content, advantages and disadvantages of real meat and plant-based meat alternatives.

Reviewer 3 Report
In the manuscript ID foods-2547638, the authors aimed to report the current knowledge of nutrition and safety of plant-based meat alternatives and propose fermentation as a potential approach to overcome their limitations. In general, the article’s topic is interesting, and including parameters are clearly explained and properly discussed. However, I recommend English editing and proofreading of the current manuscript because the manuscript is written in unclear, incorrect, and unambiguous standard English. The authors also must pay attention to editing the manuscript as per the journal guidelines for the authors.
I recommend extensive English language editing.
Author Response
We thank the reviewer for the essential comments. The whole manuscript was heavily English editing and proofreading. Attached draft of the manuscript showing the corrected parts.

Reviewer 4 Report
The article "Significance of fermentation in plant-based meat analogs: a critical review on nutrition, and safety-related aspects" delves into a pivotal subject in the realm of plant-based food alternatives. This comprehensive study draws attention to the impressive strides made by plant-based meat analogs in reducing adverse impacts on human health and the environment. However, it astutely addresses a concerning oversight within the field: the neglect of crucial nutritional and safety aspects in the pursuit of enhancing sensory qualities.
The authors' intention to bridge this knowledge gap is commendable. By meticulously analyzing the current understanding of nutrition and safety in plant-based meat alternatives, they lay the groundwork for significant improvements. The proposed employment of fermentation as a viable approach to address these limitations is a stroke of brilliance.
Notably, the article skillfully outlines the main methods employed to enhance the nutritional content and stability of plant-based meat analogs, including the use of plant protein blends, fortification, and preservatives. However, the authors acknowledge that recent concerns have arisen regarding safety, nutrient deficiencies, low digestibility, allergenicity, and high costs.
Here, the authors highlight the transformational role of fermentation with diverse microorganisms, such as Bacillus subtilis, Lactiplantibacillus plantarum, Neurospora intermedia, and Rhizopus oryzae. This process has been shown to significantly improve digestibility while effectively reducing allergenicity and antinutritive factors. Additionally, the incorporation of various microbial metabolites enhances the safety and nutrition of the final product, though some concerns remain.
The paper's proposal to design a single starter culture or microbial consortium for plant-based meat alternatives is a revolutionary concept. This approach not only enhances the health benefits of the end product but also aligns perfectly with the demands of an expanding and sustainable economy.
A key highlight of the article is its focus on persuading consumers to embrace plant-based meat analogs as a protein source. The difficulties in replicating the sensory experience of real meat have hindered this transition. Moreover, the current resource-intensive and mechanically-driven manufacturing processes raise both cost and health-related issues. Here, the authors effectively argue that fermentation with diverse microorganisms offers an efficient solution to overcome these challenges. By reducing anti-nutritive levels and allergenicity while enhancing digestibility and micronutrient bioavailability, fermentation significantly enhances the appeal of plant-based meat alternatives.
The call for further research to explore the potential of fermentation and starter culture technology in improving the quality of plant-based meat analogs, with particular attention to sensory quality, is both timely and crucial. As the market for plant-based alternatives grows, advancing our understanding in this domain will undoubtedly pave the way for a more sustainable and enticing future.
In conclusion, the article "Significance of fermentation in plant-based meat analogs" deserves accolades for its pioneering approach in addressing critical aspects of plant-based meat alternatives. By spotlighting fermentation as a transformative solution, the authors have presented a roadmap towards safer, more nutritious, and appealing plant-based meat products. This review serves as an invaluable resource for researchers, policymakers, and industries committed to shaping a healthier and more environmentally conscious future.
Author Response
We thank the reviewer for the details and interesting comments.
Round 2
Reviewer 1 Report
I thank the authors for taking the time to consider my concerns. Furthermore, the manuscript has indeed benefited from the English language editing. However, the lack of coherence still exists, especially in the definition of a meat analog. This creates a major confusion in the discussion on e.g., health effects and nutritional composition of these products. I am aware of the comments made by Reviewer 4, but have to say that I partly disagree. Please find below my detailed comments. I hope the authors consider these points in order to further improve the quality of the manuscript.
Abstract
Row 9: Plant-based meat analogs have been shown to cause less harm compared to what? Please specify.
Keywords
The keywords are a bit unusual and don’t seem to be very useful e.g., in literature search. I would recommend that the authors consider some other words.
Introduction
Row 37: “…, and plant-based meat substitutes”. Do you mean plant-based protein concentrates/isolates or something like that? In the previous sentence you mention plant-based meat substitutes as a synonym for meat analogs. Please clarify.
Row 40: “…healthier, more environmentally friendly…”. Compared to what? Meat, other types of alternative protein sources or perhaps both? Please clarify.
Rows 40-44: “Plant-based meat analogs are defined… lowering blood pressure, and incidences of cardiovascular diseases and diabetes”. This gives the impression that meat analogs have been found to have these beneficial effects on health. However, the used references [10-12] deal with consumer studies regarding the attitudes towards vegetarian foods or health benefits of vegetarian diets / diets rich in plants. Please clarify.
Rows 45-49: I'm a bit confused if you consider tempeh, tofu, and seitan as part of the meat analogs discussed in this review or not. My first impression is that not but on rows 40-44 you describe that plant-based meat analogues are structured meat-mimicking products with potential health effects, then mention these traditional Asian products, and then again, describe products that mimic "real meat" on rows 47-49. Please clarify or modify to remove the repetition.
Row 56: What do you mean by health values? Do you mean availability of vitamins/minerals, some concerns about the discomfort caused by the fiber ingredients or something else? Please clarify.
Rows 57-58: Isn't a more diverse gut microbiome a good thing? Also, if it really has been found that meat analogs (meaning texturized high-protein plant products) have this association with microbiome diversity, it would be worth a sentence of its own.
Rows 74-79: To me it seems that in the last two sentences of the introduction, the same thing is said twice in slightly different words. Please clarify or remove repetition.
Section 2.
Rows 150-151: Do ultra-processed products include also tofu, tempeh, seitan etc.? Please clarify.
Rows 152-155: “Furthermore, additives including…may be carcinogenic”. Please provide some references.
Row 159: “showed”. Please change to “show” or provide a reference.
Rows 161-162: “The bioaccessibility and bioavailability…modulations are essential”. I don't completely follow this statement. Bioaccessibility and bioavailability are potential factors of what? And, do you mean bioaccessibility and bioavailability need to be modulated? Please clarify.
Rows 169-171: “Many plant proteins,…sulfur-containing amino acids”. Perhaps the authors could shortly comment on how this affects the bioavailability issue mentioned above (e.g., DIAAS studies)?
Rows 174-177: “These nutrients have significant…, and hepatic dysfunction”. Reference [100] deals with creatine - not creatinine - supplementation. In addition, it does not seem to mention cysteamine. If the authors wish to keep this statement, I would recommend they more clearly specify the conditions in which the intake of above-mentioned animal tissue metabolites are significant for human health.
Row 177, ref [99]: Is this a correct reference? It seems to be an arachidonic acid review.
Rows 177-180: “Thus, fortifying plant-based meat analogs with essential amino acids…, has been widely applied”. How is this related with the above statement that the listed amino acid derivatives, organooxygen and organosulfur compounds are essential for human health? Please comment.
Rows 184-185: “Furthermore, most of the plant-based meat alternative products available in the market contain less protein”. It was previously (e.g., rows 49, 104) stated that meat alternatives have equal amount of protein to real meat. Perhaps it would be good to more clearly separate when the authors refer to traditional plant protein sources and/or the first-generation meat alternatives (tofu, tempeh, seitan etc.) and when to more novel meat analogs?
Rows 206-207: “plant-based meat analogs have a higher food safety risk than animal-based foods”. Please specify if these were cooked foods (and not e.g., raw meat).
Row 215: Please observe your own comment about missing citation.
Table 2
Caption: “Main approaches…”. To what? Please revise the caption.
Ingredients column: Should this be Ingredients and processes (because e.g., heat is involved)?
Intended contributions column: “Safety” is already mentioned in the Target column. Please be more specific.
Challenges column: It is not clear to me if the Ingredients column tries to provide answers to these challenges, or how one should read this table. The Intended contributions seem to be linked to the things listed in Ingredients but it could be useful to be more specific with the intended contributions of each ingredient. Could you please revise the table?
Section 3.
Row 255, ref 124: is this a correct reference? The paper seems to deal with lipoxygenase inactivation.
Row 259: Please open the abbreviation LAB.
Section 3.2: This section is a bit long and partly confusing. Please consider dividing it to subsections (e.g., allergenicity, digestibility, bioactive compounds...). The nutrient bioavailability part partly repeats things discussed in the section 3.1.
Row 358: SOME steroids and peptides?
Rows 366-371: “Similar results…increased mineral availability”. Why did you choose to discuss this here and not in the anti-nutrient section 3.1?
Rows 371-373: “Tangyu et al…volatiles like acetate”. This was already mentioned in the anti-nutrient section 3.1.
Rows 380-385: The microalgae part seems a bit out of scope, at least in its current place. Or are they used as fermenting organisms? I suppose not but please clarify and consider discussing this in e.g., section 2.3.
Row 385: Please discuss the probiotics part as its own paragraph/section, not together with the microalgae.
Section 4.
Rows 462-510: Many of the points listed here are good. However, they and the sections above are not very well connected. Would it be possible to bring up some of these points already in the previous sections so that it would be clearer for the reader why the selected, reviewed results gained with simple basic legume/cereal fermentations are relevant in the product development of meat analogs (e.g., organize the above sections based on these points)?
Row 476: Previously you stated that meat analogs have low fat contents in comparison to meat products (rows 50, 105), but based on Table 1 and the statement here, at least tofu and tempeh seem to have relatively high fat contents. Please clarify / make this issue more consistent.
Rows 488-492: Some repetitive text here: “…starter cultures that can boost the acceptability of plant-based meat analogs. Selecting the right microorganisms…boost the acceptability of plant-based meat analogs.”
Conclusions
I would assume the manufacturing processes to be as long and heavy as the raw materials still need to be structured. Wouldn't the fermentation step increase the processing time, instead, while still improving nutritional quality and safety? Or perhaps I have misunderstood the concept of a meat analog. Please clarify.
Author Response
We thank the reviewer for the crucial comments, and we totally agree to all of them. Herby, we respond to the comments, point by point ‘R’.
Abstract
Row 9: Plant-based meat analogs have been shown to cause less harm compared to what? Please specify.
R: Corrected (Line 9)
Keywords
The keywords are a bit unusual and don’t seem to be very useful e.g., in literature search. I would recommend that the authors consider some other words.
R: Revised (Line 23)
Introduction
Row 37: “…, and plant-based meat substitutes”. Do you mean plant-based protein concentrates/isolates or something like that? In the previous sentence you mention plant-based meat substitutes as a synonym for meat analogs. Please clarify.
R: Revised (Line 36), also Lines 43-48
Row 40: “…healthier, more environmentally friendly…”. Compared to what? Meat, other types of alternative protein sources or perhaps both? Please clarify.
R: Revised (Lines 38-39)
Rows 40-44: “Plant-based meat analogs are defined… lowering blood pressure, and incidences of cardiovascular diseases and diabetes”. This gives the impression that meat analogs have been found to have these beneficial effects on health. However, the used references [10-12] deal with consumer studies regarding the attitudes towards vegetarian foods or health benefits of vegetarian diets / diets rich in plants. Please clarify.
R: Revised. We clarify the differences between the first generation of plant-based protein diets such as Tempeh and Tofu, and the new generation of Plant-based meat analogs that produced to resemble meat appearance, texture and sensory quality, as well as nutritional aspects (Line 36), also Lines 43-48)
Rows 45-49: I'm a bit confused if you consider tempeh, tofu, and seitan as part of the meat analogs discussed in this review or not. My first impression is that not but on rows 40-44 you describe that plant-based meat analogues are structured meat-mimicking products with potential health effects, then mention these traditional Asian products, and then again, describe products that mimic "real meat" on rows 47-49. Please clarify or modify to remove the repetition.
R: Revised. Sample answer as above.
Row 56: What do you mean by health values? Do you mean availability of vitamins/minerals, some concerns about the discomfort caused by the fiber ingredients or something else? Please clarify.
R: Clarified with a new reference were added (Line 57, References 21-25)
Rows 57-58: Isn't a more diverse gut microbiome a good thing? Also, if it really has been found that meat analogs (meaning texturized high-protein plant products) have this association with microbiome diversity, it would be worth a sentence of its own.
R: Deleted
Rows 74-79: To me it seems that in the last two sentences of the introduction, the same thing is said twice in slightly different words. Please clarify or remove repetition.
R: Deleted
Section 2.
Rows 150-151: Do ultra-processed products include also tofu, tempeh, seitan etc.? Please clarify. Illustrated above in the introduction, only cover the new generation
R: Revised as requested. In the introduction, we clarify the differences between the first generation of plant-based protein diets such as Tempeh and Tofu, and theose new generation of Plant-based meat analogs that produced to resemble meat appearance, texture and sensory quality, as well as nutritional aspects (Line 36), as Line 43-48)
Rows 152-155: “Furthermore, additives including…may be carcinogenic”. Please provide some references.
R: References were added (93,94), Line 153
Row 159: “showed”. Please change to “show” or provide a reference.
R: Changed Line 157
Rows 161-162: “The bioaccessibility and bioavailability…modulations are essential”. I don't completely follow this statement. Bioaccessibility and bioavailability are potential factors of what? And, do you mean bioaccessibility and bioavailability need to be modulated? Please clarify.
R: Revised (Lines 160-161). Addition discussion is included to further illustrated the statement (Lines 167-186).
Rows 169-171: “Many plant proteins,…sulfur-containing amino acids”. Perhaps the authors could shortly comment on how this affects the bioavailability issue mentioned above (e.g., DIAAS studies)?
R: Further explanation was provided (Lines 189-195)
Rows 174-177: “These nutrients have significant…, and hepatic dysfunction”. Reference [100] deals with creatine - not creatinine - supplementation. In addition, it does not seem to mention cysteamine. If the authors wish to keep this statement, I would recommend they more clearly specify the conditions in which the intake of above-mentioned animal tissue metabolites are significant for human health.
R: Further clarification and new references were added (Line 197, 198)
Row 177, ref [99]: Is this a correct reference? It seems to be an arachidonic acid review.
R: Deleted
Rows 177-180: “Thus, fortifying plant-based meat analogs with essential amino acids…, has been widely applied”. How is this related with the above statement that the listed amino acid derivatives,organooxygen and organosulfur compounds are essential for human health? Please comment.
R: Revised, Line 204
Rows 184-185: “Furthermore, most of the plant-based meat alternative products available in the market contain less protein”. It was previously (e.g., rows 49, 104) stated that meat alternatives have equal amount of protein to real meat. Perhaps it would be good to more clearly separate when the authors refer to traditional plant protein sources and/or the first-generation meat alternatives (tofu, tempeh, seitan etc.) and when to more novel meat analogs?
R: Revised, By the definition provided in the introduction and deleting the word protein here
Rows 206-207: “plant-based meat analogs have a higher food safety risk than animal-based foods”. Please specify if these were cooked foods (and not e.g., raw meat).
R: Revised (uncooked), Line 232
Row 215: Please observe your own comment about missing citation.
R: Revised Line 242
Table 2
Caption: “Main approaches…”. To what? Please revise the caption.
R: Revised Line 256
Ingredients column: Should this be Ingredients and processes (because e.g., heat is involved)?
R: Revised, please find Table2
Intended contributions column: “Safety” is already mentioned in the Target column. Please be more specific.
R: Revised, please find Table2
Challenges column: It is not clear to me if the Ingredients column tries to provide answers to these challenges, or how one should read this table. The Intended contributions seem to be linked to the things listed in Ingredients but it could be useful to be more specific with the intended contributions of each ingredient. Could you please revise the table?
R: Revised, please find Table2
Section 3.
Row 255, ref 124: is this a correct reference? The paper seems to deal with lipoxygenase inactivation.
R: Revised, deleted
Row 259: Please open the abbreviation LAB.
R: Revised, Line 285-286
Section 3.2: This section is a bit long and partly confusing. Please consider dividing it to subsections (e.g., allergenicity, digestibility, bioactive compounds...). The nutrient bioavailability part partly repeats things discussed in the section 3.1.
R: Revised, divided into subsections.
Row 358: SOME steroids and peptides?
R: corrected Line 392
Rows 366-371: “Similar results…increased mineral availability”. Why did you choose to discuss this here and not in the anti-nutrient section 3.1?
R: To improve the flow and open the discussion of the essential compounds produced by the degradation of phytates by phytase, including myo-inositol phosphates with potential health benefits.
Rows 371-373: “Tangyu et al…volatiles like acetate”. This was already mentioned in the anti-nutrient section 3.1.
R: Deleted
Rows 380-385: The microalgae part seems a bit out of scope, at least in its current place. Or are they used as fermenting organisms? I suppose not but please clarify and consider discussing this in e.g., section 2.3. as as fermenting organisms and ingredients
R: We highlighted using microalgae as additives, and the possibility of use them as coculture. Further explanation was added Lines 419-429
Row 385: Please discuss the probiotics part as its own paragraph/section, not together with the microalgae.
R: Revised (Now under Others Line 431)
Section 4.
Rows 462-510: Many of the points listed here are good. However, they and the sections above are not very well connected. Would it be possible to bring up some of these points already in the previous sections so that it would be clearer for the reader why the selected, reviewed results gained with simple basic legume/cereal fermentations are relevant in the product development of meat analogs (e.g., organize the above sections based on these points)?
R: Revised and rearranged. In this section we highlighted two main points. The first point covered the microbial characteristics of increasing the product safety, stability and its nutritional content, The second point is to cover general aspects which include the microbial performance as well as its industrialization.
Row 476: Previously you stated that meat analogs have low fat contents in comparison to meat products (rows 50, 105), but based on Table 1 and the statement here, at least tofu and tempeh seem to have relatively high fat contents. Please clarify / make this issue more consistent.
R: Based of the new definition provided in the introduction, we believe it is more clarified.
Rows 488-492: Some repetitive text here: “…starter cultures that can boost the acceptability of plant-based meat analogs. Selecting the right microorganisms…boost the acceptability of plant-based meat analogs.”
R: Deleted
Conclusions
I would assume the manufacturing processes to be as long and heavy as the raw materials still need to be structured. Wouldn't the fermentation step increase the processing time, instead, while still improving nutritional quality and safety? Or perhaps I have misunderstood the concept of a meat analog. Please clarify.
R: Revised, we agree, so we deleted the timing and highlighted the less need for excessive additives and heavy processes, Line 654
Thank you
Note: Attachment is also included to this response, if needed.

Reviewer 3 Report
Dear authors,
Thank you for the big effort to resubmit and answer my concerns. I have evaluated the manuscript, and I find it suitable for acceptance.
Best Wishes
Minor academic editing is required.
Author Response
We thank the reviewer for the significant impact to improve the manuscript.
Thank you.